# Catalytic asymmetric [4 + 2] dearomative photocycloadditions of anthracene and its derivatives with alkenylazaarenes

Dong Tian [1], Wenshuo Shi[1], Xin Sun [1]✉, Xiaowei Zhao[1], Yanli Yin [2,3]✉ & Zhiyong Jiang [1,2]✉

Photocatalysis through energy transfer has been investigated for the facilitation of [4 + 2] cycloaddition reactions. However, the high reactivity of radical species poses a challenging obstacle to achieving enantiocontrol with chiral catalysts, as no enantioselective examples have been reported thus far. Here, we present the development of catalytic asymmetric [4 + 2] dearomative photocycloaddition involving anthracene and its derivatives with alkenylazaarenes. This accomplishment is achieved by utilizing a cooperative photosensitizer and chiral Brønsted acid catalysis platform. Importantly, this process enables the activation of anthracene substrates through energy transfer from triplet DPZ, thereby initiating a precise and stereoselective sequential transformation. The significance of our work is highlighted by the synthesis of a diverse range of pharmaceutical valuable cycloadducts incorporating attractive azaarenes, all obtained with high yields, ees, and drs. The broad substrate scope is further underscored by successful construction of all-carbon quaternary stereocenters and diverse adjacent stereocenters.

Cycloaddition is an essential and versatile tool to construct diverse (hetero)cycles[1,2]. The most widely adopted methods include [2 + 2], [3 + 2], and [4 + 2] cyclization reactions, wherein photo-activation or energy transfer-based photocatalysis[3,4] has been widely used to accomplish [2 + 2] transformations of 2π systems with other 2π systems[5–25] or 2σ species[26–28] (Fig. 1A). The rapid advance of [2 + 2] photocycladditon, in particular, has been highlighted by the establishment of a significant number of enantioselective manifolds by incorporating extrinsic chiral catalysts in reaction systems, presenting valuable enantioenriched cyclobutanes. Conversely, in terms of the principle of conservation of orbital symmetry[29], [4 + 2] cycloaddition is favored to occur via the ground-state ionic pathway, with the Diels-Alder reactions being well-known representatives (Fig. 1A)[1,2,30]. The importance of cyclohexane derivatives has motivated chemists to devise countless asymmetric catalytic strategies over the past decades. Nonetheless, the pursuit of employing simple yet inert feedstocks

under mild reaction conditions and with broad functional group tolerances still remains a highly desirable task in organic synthesis, with considerable implications for extending the methodology to the pharmaceutical industry. To this end, several visible light-driven energy transfer-enabled[31–34] [4 + 2] photocycloaddition reactions have been explored by the groups of Glorius, Maji, and You et al. (Fig. 1A)[35–41]. Notably, despite the potential benefits of this sustainable and versatile catalytic platform, which leverages the high reactivity of the triplet species and entails multiple steps involving radical addition, intersystem crossing (ISC), and radical coupling, no instances of enantioselective manifolds have been documented to date. Challenges in achieving enantioselectivity in these reactions stem from the difficulty faced by chiral catalysts in providing sufficient enantiofacial differentiation for the formation of stereocenters. Additionally, the intrinsic racemic background reactions pose further hurdles to achieving enantioselectivity[42–46].

[1]Key Laboratory of Natural Medicine and Immuno-Engineering of Henan Province, Henan University, Jinming Campus, Kaifeng, Henan, P. R. China. [2]School of Chemistry and Chemical Engineering, Henan Normal University, Xinxiang, Henan, P. R. China. [3]College of Advanced Interdisciplinary Science and Technology, Henan University of Technology, Zhengzhou, Henan, P. R. China. ✉e-mail: sunxin0910@qq.com; yinzihust@163.com; jiangzhiyong@htu.edu.cn

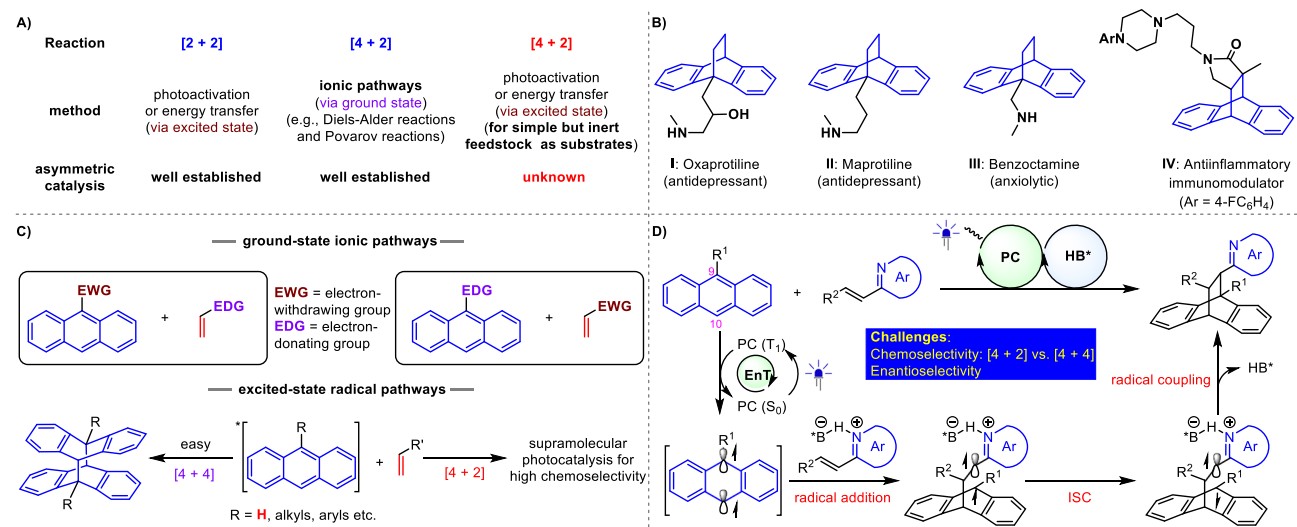

**Fig. 1 | Outline of this work. A** Background of enantioselective [2 + 2] and [4 + 2] cycloaddition reactions. **B** Representative biologically active molecules. **C** [4 + 2] cycloaddition of anthracene and its derivatives. **D** Design plan.

In recent years, we have devoted tremendous effort to the development of various useful asymmetric photocatalytic reactions driven by visible light[47]. The investigations have revealed that the dual catalyst system combining a photosensitizer and a chiral Brønsted acid is a robust platform in the preparation of enantiomerically pure imine-containing azaarene variants[48–57]. The centrality to the success of this catalysis platform stems from the high reactivity of radicals, which allows the applicability of relatively simple azaarene-based substrates, even though these azaarenes feature poor electron-withdrawing ability[58]. Moreover, the basicity of the nitrogen atom of azaarenes leads to a preferential and solid hydrogen-bonding (H-bonding) interaction between the azaarene-based substrate and the chiral Brønsted acid catalysts, thereby providing an efficient enantiocontrol environment. Motivated by these findings, we were intrigued to explore the unmet [4 + 2] photocycloaddition reaction of alkenyla-zaarenes and, more importantly, to challenge the enantioselective manifold. This endeavor is poised to facilitate the rapid synthesis of promising cyclohexane derivatives functionalized by azaarenes that are ubiquitous in pharmaceuticals and biologically active natural and non-natural products[59].

Among the established [4 + 2] cycloaddition reactions, anthracene derivatives are of particular synthetic interest owing to the convenient availability of these feedstocks and the attractive bioactivities of the resulting cycloadducts (e.g., molecules **I-IV**, Fig. 1B)[60–63]. In order to break the inherent aromatic stability of anthracene, activated functional groups are typically required to assemble on the anthracene ring (Fig. 1C)[64]. For simple anthracene derivatives and diverse reaction partners with similar poor reactivity to be viable and for conversion to occur under mild reaction conditions, the exploitation of photocatalytic systems to input additional chemical energy from photons has been investigated by chemists[65]. But it has been demonstrated that triplet anthracene is also susceptible to undergoing [4 + 4] dimerization[39,65]. To exclusively yield [4 + 2] cycloadducts, supramolecular photocatalysis via charge-transfer excitation within host-guest complexes had to be devised[38]. Given the electrophilic nature of alkenylazaarenes, we speculated that the radical at the 10-position of triplet anthracene with less steric hindrance might be prone to undergoing addition reactions due to its nucleophilic capacity. Meanwhile, the resulting electrophilic α-azaaryl radical can experience ISC to further couple with the radical at the 9-position of anthracene. In this context, the exploration of photocatalytic asymmetric [4 + 2] dearomative

photocycloadditions of anthracene and its derivatives with vinyla-zaarenes became our desired research objective.

## Results

To probe the viability of this tentative scenario, the simplest anthracene (**1a**) and 2-(1-phenylvinyl)quinoline (**2a**) were selected as the model substrates (Table 1 and Supplementary Table 1). The choice of **2a** stems from the following considerations: 1) terminal olefins should possess higher reactivity than their corresponding non-terminal variants to facilitate the addition of radicals, owing to less steric hindrance, making it useful for evaluating the feasibility of the reaction; 2) the formation of all-carbon quaternary stereocenters substituted by azaarenes from simple azaarene-based substrates always represents an important but challenging task[50]. The transformation was initially examined at 25 and even 60 °C in the absence and presence of 20 mol% diphenyl phosphate as the Brønsted acid catalyst and in toluene as the solvent. As expected, no reaction was observed, verifying the inapplicability of thermal conversion. Given that the triplet energy of **1a** is 41.5 kcal/mol, a dicyanopyrazine chromophore (DPZ, $E_T$ = 46.4 kcal/mol)[66] as the visible light-absorbing sensitizer was attempted as the photocatalyst. We were pleased to find that the desired product **3a** was obtained in 39% yield in the presence of 0.5 mol% DPZ and irradiated by a 3 W blue LED. This encouraged us to explore a series of chiral Brønsted acids and other reaction parameters for the enantioselective manifold (Supplementary Table 1). Consequently, at −35 °C with irradiation by three 3 W blue LEDs, **3a** could be achieved in 83% yield with 90% ee when employing 10 mol% (S)-SPINOL-derived N-triflyl-phosphoramide **C1** in dichloromethane as the solvent within 36 h (entry 1, Table 1). Other photosensitizers were then evaluated, such as Eosin Y ($E_T$ = 39.7 kcal/mol)[32], Rose Bengal ($E_T$ = 40.9 kcal/mol)[33], 4CzIPN ($E_T$ = 57.1 kcal/mol)[32], Ir(ppy)$_3$ ($E_T$ = 57.8 kcal/mol)[33] and (Ir[dF(CF$_3$)ppy]$_2$(dtbbpy))PF$_6$ ($E_T$ = 60.8 kcal/mol)[33] (entries 2 – 6). The results clearly show that the photosensitizers with higher triplet energy (entries 4 – 6), but not lower (entries 2 – 3), than **1a** can allow for a smooth conversion. It should be noted that only a 45% yield of **3a** was obtained in the presence of Ir(ppy)$_3$ as a photocatalyst, which originates from poor reactivity, most likely due to insufficient energy transfer efficiency through molecular collisions (entry 5). Indeed, when extending the reaction time to 96 h, the transformation could be completed. Chiral N-triflyl-phosphoramides **C2** and **C3** featuring other substituents at the 6,6-positions of SPINOL, were then examined, but no better results could be obtained (entries 7 – 8). The transformation

**Table 1 | Optimization of the Reaction Conditions[a]**

| entry | alteration to conditions | yield (%)[b] | ee (%)[c] |
|---|---|---|---|
| 1 | none | 83 | 90 |
| 2 | Eosin Y instead of DPZ | N.R.[d] | N.A. |
| 3 | Rose Bengal instead of DPZ | N.R.[d] | N.A. |
| 4 | 4CzIPN instead of DPZ | 82 | 89 |
| 5 | Ir(ppy)$_3$ instead of DPZ | 45 | 90 |
| 6 | Ir$^{III}$ complex[e] instead of DPZ | 87 | 89 |
| 7 | **C2** instead of **C1** | 57 | 75 |
| 8 | **C3** instead of **C1** | 31 | 58 |
| 9 | no **C1** | 12 | N.A. |
| 10 | no DPZ | N.R. | N.A. |
| 11 | no light | N.R. | N.A. |
| 12 | under air | N.P. | N.A. |

[a]Reaction conditions: **1a** (0.10 mmol), **2a** (0.12 mmol), DPZ (5.0 × 10$^{-4}$ mmol) and **C1** (0.01 mmol) in degassed DCM (5.0 mL) and at −35 °C. [b]Yield of isolated product. [c]Ees were determined by HPLC analysis. [d]After 96 h, trace product **3a** was observed. [e]Ir$^{III}$ complex = (Ir[dF(CF$_3$)ppy]$_2$(dtbbpy))PF$_6$. N.R. = no reaction. N.A. = not applicable. N.P. = no product. 4CzIPN = 2,4,5,6-tetrakis(carbazol-9-yl) −1,3-dicyanobenzene. ppy = 2-(2-pyridyl)phenyl. dF(CF$_3$)ppy = 2-(2,4-difluorophenyl)−5-trifluoromethylpyridine. dtbbpy = 4,4-di-tert-butyl-2,2′-bipyridine.

was then performed without **C1**, resulting in **3a** in only 12% yield originating from the considerably worse chemical conversion (entry 9), suggesting that lowering the temperature could effectively suppress the racemic background reaction. Notably, subsequent control experiments revealed that DPZ, visible light, and oxygen-free environments are indispensable for the transformation to occur (entries 10 – 12), wherein **1a** was exhausted, but no **3a** was detected when in the presence of an ambient atmosphere. It is worth mentioning that no [4 + 4] photocycloaddition product of **1a** was detected in these reactions, suggesting that the current photocatalysis platform is not suitable for such a competitive transformation. With optimized conditions in hand, the scope of this asymmetric [4 + 2] dearomative photocycloaddition reaction was explored (Fig. 2). We initially attempted anthracene (**1a**) to react with 2-(1-arylvinyl)quinolines **2** containing various electron-withdrawing and electron-donating groups at distinct positions on aromatic rings. It was observed that the corresponding products **3b–k** were obtained in 67 to 95% yields with 84 to 92% ee. The method demonstrated a robust ability to assemble all-carbon

quaternary stereocenters, as evidenced by the good compatibility of 2-(1-alkylvinyl)quinolines (e.g., product **3 l**), allowing the introduction of not only aryls but also alkyls at the stereocenter. Such an important advantage for organic synthesis could be successfully expanded to olefins activated by pyridines; as a representative example, products **3m-n** were achieved in 59 to 72% yields with 85 to 93% ee. To introduce useful functional groups on the aromatic rings of the 9,10-dihydroanthracene moiety of the products, the reactions of different 1,8-disubstituted anthracenes with **2a** were attempted, leading to adducts **3o-q** in 70–87% yields with 90–97% ee. It is worth mentioning that the regioselectivity is excellent (>20:1 rr), motivating further evaluation of 1-substituted anthracenes. We found that the reaction between 1-chloroanthrancene and **2a** resulted in product **3r** in 70% yield with 89%/88% ee, 2.5:1 dr and >20:1 rr. Although the dr value is not satisfactory, the effect of chlorine as a substituent on diastereoselectivity and regioselectivity is clearly revealed.

In asymmetric catalysis, the construction of two adjacent all-carbon quaternary stereocenters is of importance but poses a formidable challenge. In this context, a series of 9-substituted anthracenes were subjected to reactions with diverse α-aryl and -alkyl vinylazaarenes, even although the newly formed quaternary carbon on the 9-position of the 9,10-dihydroanthracene moiety is not a real stereocenter. As a result, products **3s-x** were obtained in high yields and ees. Importantly, the method exhibited a broad substrate scope, tolerating anthracenes with various electron-donating or withdrawing groups on the 9-position, such as methylalcohol (**3s-t**), ketones (**3 u**), halides (**3 v**) and alkyl ethers (**3x**). Azaarene-substituted ethylenes were then explored, furnishing adducts **3y-zc** in 43 to 92% yields with 66 to 90% ee. Finally, we applied the method to assemble bioactive molecules, especially drugs, on such an attractive bridged cyclic product, effectively serving the purpose of drug discovery. We were pleased to find that products **3zd-zg** were obtained with satisfactory results, introducing pharmaceutical molecules 2,4-D (**3zd**), probenecid (**3ze**), adapalene (**3zf**), and gabapentin (**3zg**) on diverse positions of the products. It is worth mentioning that careful modulation of the reaction parameters, including chiral catalyst, temperature, and solvent, is required to achieve high enantioselectivity, highlighting the elusive challenge of enantioselectivity in such photocycloaddition reactions, which are highly sensitive to these factors.

Encouraged by this success, we continued to explore the viability of alkenyl azaarenes (i.e., β-substituted vinylazaarenes, Fig. 3). However, no reaction was observed for the transformations of anthracene (**1a**) with diverse alkenes (**4**). This dilemma prompted us to test 9-substituted anthracenes, considering that the substituent may affect the reactivity of radical species. Finally, anthracen-9-ylmethanol (**1 f**) was determined as the suitable substrate, as the transformation of (E) −4-chloro-2-styrylquinoline (**4a**) with it could render product **5a** in 92% yield with 92% ee and >20:1 dr when using (S)-H$_8$−1,1′-Bi-2-naphthol (BINOL)-based chiral phosphoric acid (CPA) **C6** as the chiral catalyst. A series of (E)−2-styrylquinolines containing diverse substituents on either quinolyl rings or 2-aromatic rings of olefins were compatible

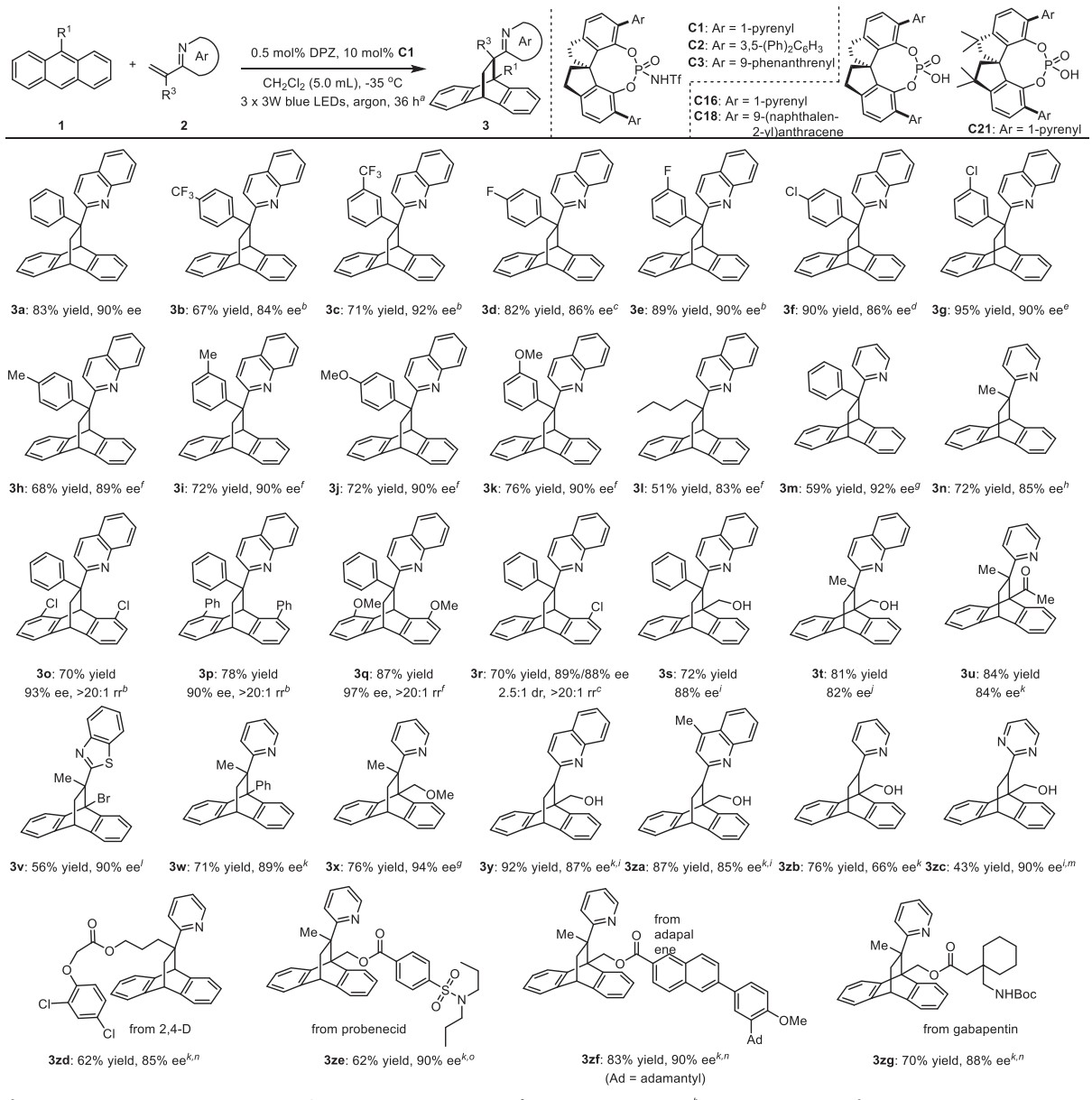

**Fig. 2 | Asymmetric [4 + 2] photocycloadditions of anthracene and its derivatives with vinylazaarenes.**

well, resulting in products **5b-o** in 55 to 98% yields with 86 to 96% ee and >20:1 dr. Further study revealed that such olefins with 2-pyridyl as the activating group were also the feasible reaction partners of **1 f**. However, no reaction was observed for the β-alkyl substituted vinylazaarenes, revealing a remaining challenge. It is worth mentioning that the effect of methyl alcohol on reactivity is not clear. Nonetheless, we considered that this might stem from the ability of the hydroxyl group to provide an H-bonding donor to interact with O = P of the chiral catalyst, slightly reducing the activation energy to trigger these poorer reactive transformations originating from the higher steric hindrance of **4** compared to terminal olefins **2** and the weaker electron-deficient nature of **4** compared to 1,3-diketones **6** (*vide infra*). This speculation is supported by the result that no reaction occurred when OH was modified as OMe.

In order to further explore the viability of this catalysis platform to furnish a more diverse variety of complex and enantioenriched azaarene-functionalized dearomative cycloadducts from

anthracenes, we proceeded to test azaarene-activated 1,3-diketones **6** as the partners, with the enol form being the dominant isomer (Fig. 4). Given the strong electron-withdrawing ability of ketones and the less steric hindrance of the ketone side of the enol form, we speculated that these entities should be workable and with excellent regioselectivity. As expected, in the presence of 1.0 mol% DPZ and 20 mol% SPINOL-CPA **C16** at −50 °C, the transformations of anthracene (**1a**) with a series of 1-azaaryl-3-aryl and −3-alkyl diketones **6** resulted in adducts **7a–j** in 81 to 99% yields with 85 to 99% ee and >20:1 dr. The less active 9-substituted anthracenes were then evaluated, and adducts **7k–l** with satisfactory yields, ees, and drs further support the robust capability of this method. Due to the rather low ratio of enol to ketone isomers, 1-azaaryl-substituted 1,3-ketoesters are not compatible with the catalysis platform. In addition, it was found that CPA **C22**, featuring the 2-phenanthryl substituents at the 6,6'-positions of SPINOL, is also capable of providing sufficient enantiocontrol in a small number of cases (i.e., **7a–c**, **7i**

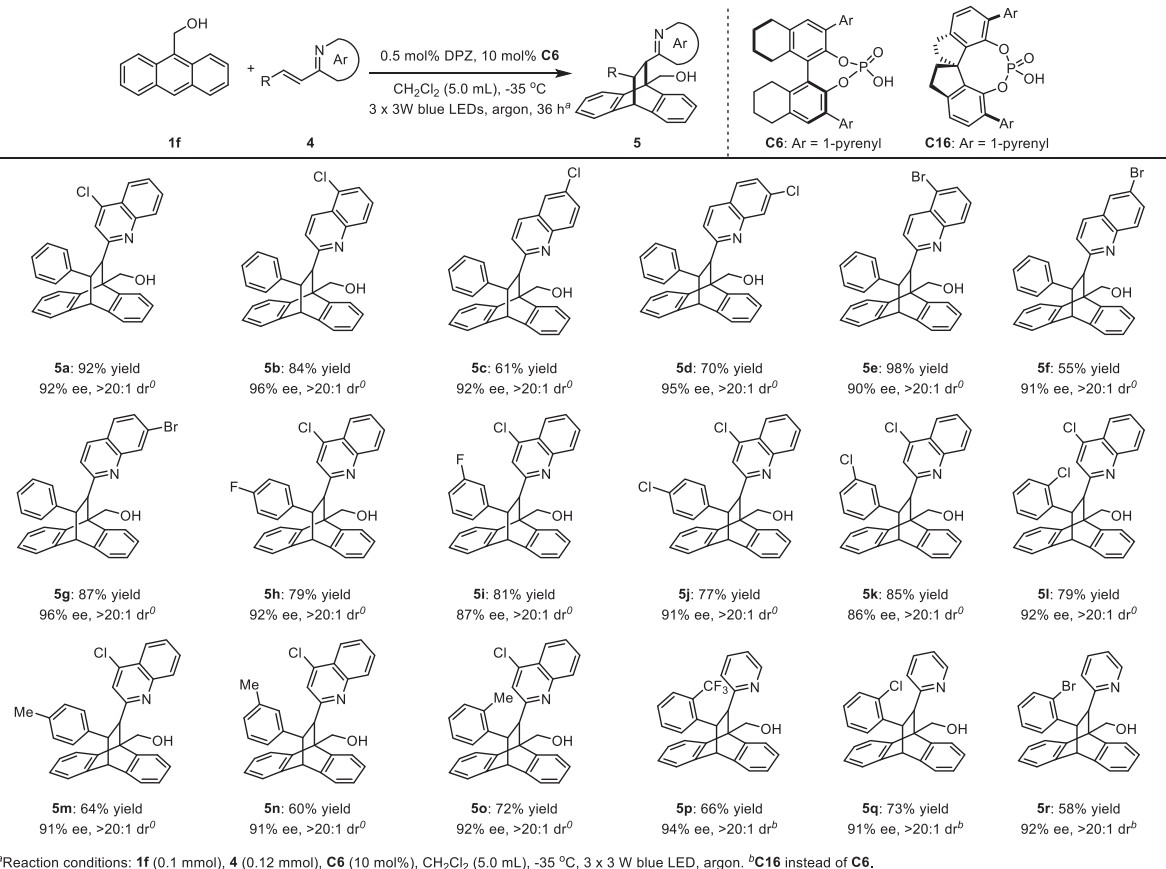

**Fig. 3 | Asymmetric [4 + 2] photocycloadditions of anthracene and its derivatives with β-substituted vinylazaarenes.**

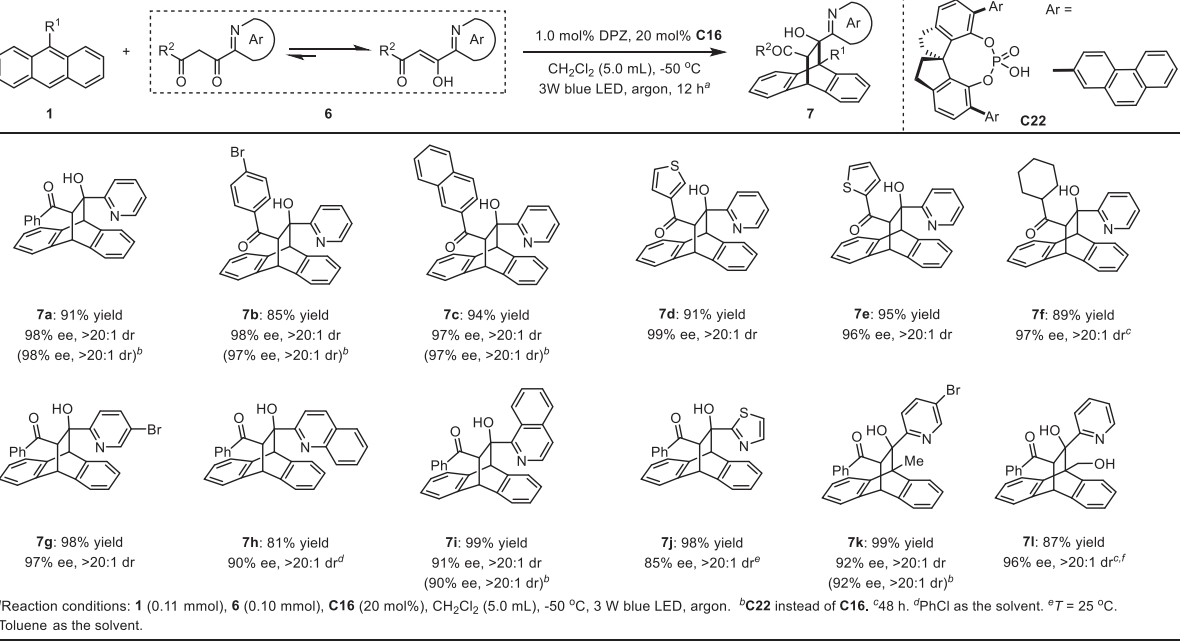

**Fig. 4 | Asymmetric [4 + 2] photocycloadditions of anthracene and its derivatives with azaarene-based 1,3-diketones.**

and **7k**), where the reactivity is similar to that of **C16** engaged in the transformations. It is worth mentioning that no reaction was observed when the azaaryl group of **6** was replaced by the simple aryl group, most likely due to the small amount of enol species in the reaction system resulting in the rather poor reactivity.

As noted above, [4 + 2] cycloaddition adducts of anthracenes are valuable to the pharmaceutical industry (Fig. 1B). Meanwhile, numerous drugs and natural products bear these imine-containing azaarenes. Accordingly, both sets of products may possess promising and attractive biological activities. Furthermore, the method enables the

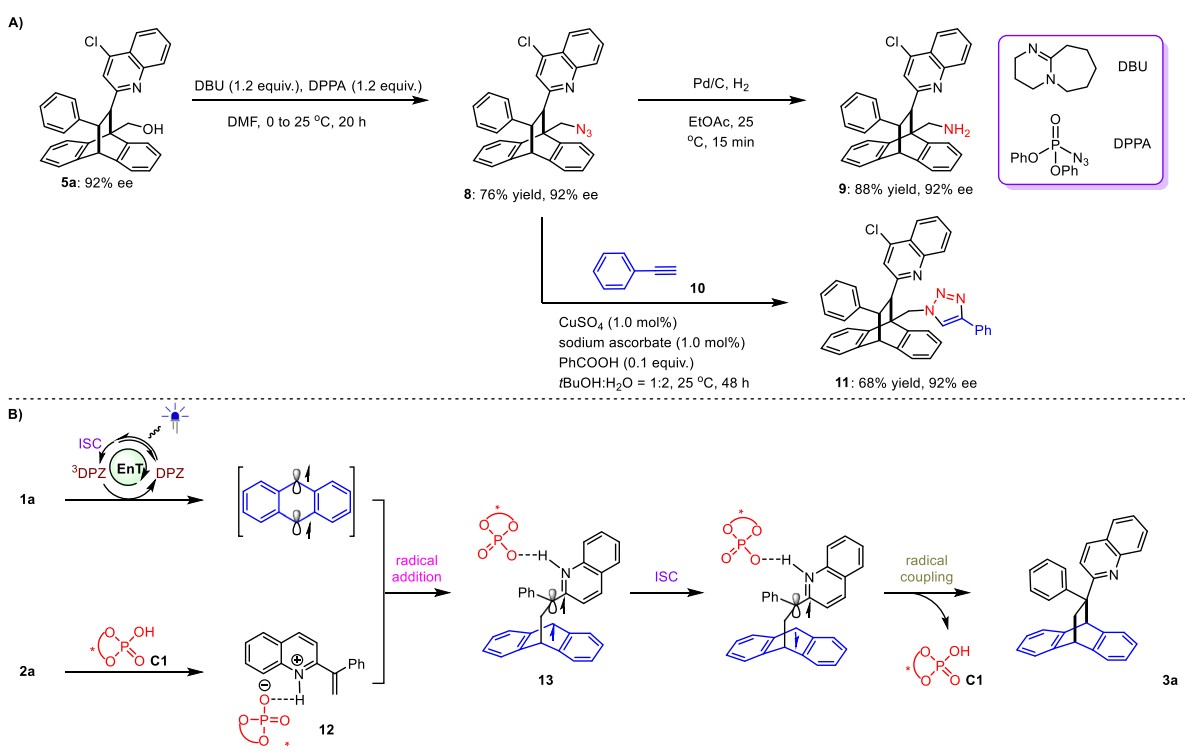

**Fig. 5 | Synthetic applications and plausible mechanism of this [4 + 2] photocycloaddition reaction.** (**A**) Synthetic utilities of products. (**B**) The proposed mechanism. ISC = intersystem crossing.

facile introduction of diverse drug molecules onto such an important enantioenriched scaffold (e.g., **3zd–zh**). As such, this asymmetric [4 + 2] photocycloaddition reaction has fascinating potential in drug discovery. It is worth noting that previous investigations have also revealed important applications of their *N*-oxide derivatives in transition-metal asymmetric catalysis as chiral ligands[67,68]. Nevertheless, simple chemical transformations of the adducts were performed to further demonstrate the synthetic application of this strategy (Fig. 5A). We selected product **5a** as a representative to undergo the Mitsunobu reaction using 1,8-diazabicyclo[5.4.0]undec-7-ene (DBU) and diphenyl azidophosphate (DPPA). Azide **8** was obtained in 76% yield with the maintained ee value. Correspondingly, in addition to the hydroxy of **5a** serving as a linker to connect the functional groups, derivative **9** generated from the reduction of **8** with Pd/C and H₂ can produce a primary amine as another versatile linker to facilitate the targeted modifications of products. To assemble the attractive triazole ring, ethynylbenzene (**10**) was attempted to react with **8**. As a result, in the presence of CuSO₄, sodium ascorbate and benzoic acid, triazole **11** was achieved in 68% yield with 92% ee.

Subsequently, we pursued the exploration of plausible mechanisms with the transformation of anthracene (**1a**) with 2-(1-phenylvinyl) quinoline (**2a**) as the model reaction. Given that no reaction occurs in the absence of DPZ (entry 10, Table 1), the direct photoexcitation mechanism is rationally excluded. Accordingly, a series of Stern–Volmer experiments with an excitation wavelength of 448 nm were conducted. As a result, no measurable luminescence quenching of the photo-excited DPZ (*DPZ) by **1a** or **2a** was observed. Therefore, a photoredox catalytic cycle is most likely not justified, further supported by the viability of Ir(ppy)₃ (entry 5, Table 1), characterized by a rather low excited-state reduction potential ($E_{1/2}(M^*/M^-) = +0.31\,V$)[33]. Moreover, the absorption spectra of both **1a** and **2a** do not overlap with the emission spectrum of DPZ, ruling out singlet-singlet EnT. We next measured the corresponding transient UV-Vis absorption spectra[69], and the results clearly demonstrate that the triplet-triplet EnT of ³DPZ and **1a** is responsible for triggering the transformation.

Given the insufficient ability of triplet anthracene ($E^t(S^{·+}/S^*) = -0.46\,V$) to reduce **2a** ($E_p = -1.08\,V$ vs SCE in CH₃CN), it is probable that triplet **1a** with a formal diradical pattern is prone to further undergo a sequential process that includes radical addition, ISC, and intramolecular radical recombination.

As depicted in Fig. 5B, the mechanism of this [4 + 2] photocycloaddition reaction is proposed, using the representative reaction between **1a** and **2a** to generate **3a**. Concerning DPZ under irradiation with visible light, the triplet species (i.e.,³DPZ) is generated after ISC. Triplet-triplet energy transfer of ³DPZ with **1a** can lead to the triplet **1a** with a diradical form. Due to the indispensability of chiral catalyst **C1** for the reaction to proceed smoothly, the formation of intermediate **12** from **2a** and **C1** via H-bonding interaction is crucial, providing an enantiocontrol environment for the subsequent radical addition. The resulting diradical species **13** will readily undergo ISC to facilitate the final radical coupling process, producing enantioenriched adduct **3a**. With respect to the enantiocontrol mode of **C1**, the fact that the polycyclic (aromatic) hydrocarbons as the substituents of CPAs (e.g., **C1–C3, C16** and **C22**) are beneficial to achieving precise enantiocontrol reveals a high probability of π-π stacking interactions between the substrates and the chiral catalyst to improve the capability of the latter to differentiate enantiofaces. Meanwhile, decreasing the temperature considerably suppresses the racemic background reactions, providing chiral catalysts the opportunity to offer enantiocontrol for the formation of the stereocenter.

## Discussion

In summary, the visible-light-driven photocatalytic asymmetric [4 + 2] cycloaddition has been addressed. The catalyst system involves a photosensitizer to perform energy transfer and a chiral Brønsted acid to provide stereocontrol. It facilitates the transformations of simple and relatively inert anthracene and its derivatives when coupled with a variety of alkenylazaarenes, all achieved efficiently under mild reaction conditions. This methodology has enabled the synthesis of a large array of dearomative cycloadducts

incorporating azaarene functionalities with high yields, ees, and drs. Notably, this approach excels in constructing distinct stereocenters, including the challenging all-carbon quaternary carbon centers, within the framework of pharmaceutically significant molecules. Furthermore, the investigation revealed that such an excited state platform is efficient in accommodating both electron-deficient and electron-rich anthracene derivatives as olefin partners, broadening the scope of applicable substrates. We anticipate that these findings will motivate the pursuit of a greater variety of asymmetric [4 + 2] photocatalytic reactions with readily accessible but valuable starting materials, thereby stimulating a rapid advance in the pharmaceutical industry.

## Methods
General experimental procedures can be found in the Supplementary Information.

## Data availability
All data are available from the authors upon request. The authors declare that all data supporting the findings of this study are available in the paper and its Supplementary Information files. The X-ray crystallographic coordinates for structures reported in this study have been deposited at the Cambridge Crystallographic Data Center (CCDC) under deposition numbers CCDC 2302299 (**3 f**), CCDC 2301849 (**3o**), CCDC 2309906 (**3 u**), CCDC 2301851 (**5k**) and CCDC 2309908 (**7 g**). These data can be obtained free of charge from The Cambridge Crystallographic Data Center via www.ccdc.cam.ac.uk/data_request/cif.

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

## Acknowledgements

Grants from the National Science Foundation of China (21925103, Z.J., 22171072, Y.Y., and 22301061, X.Z.) are gratefully acknowledged.

## Author contributions

Z.J., D.T., and X.S. conceived and designed the experiments. D.T., W.S., and X.S. performed the experiments. X.Z. and Y.Y. analyzed and interpreted the results. D.T., X.S., Y.Y., and Z.J. prepared the Supplementary Information. Z.J. and Y.Y. wrote the paper. All authors discussed the results and commented on the manuscript.

## Competing interests

The authors declare no competing interests.
