## [Peer Review File · Nature Communications]

Catalytic Asymmetric [4 + 2] Dearomative Photocycloadditions of Anthracene and Its Derivatives with AlkenylazaarenesReviewers' Comments:

Reviewer #1:

Remarks to the Author:

The direct asymmetric [4+2] cycloaddition reactions serve as a rapid and cost-effective method for creating asymmetric cyclic frameworks. In this manuscript, Jiang et al. report a photo-induced asymmetric [4+2] cycloaddition reaction by employing organic photocatalysts in tandem with chiral Brønsted acid catalysts. The process initiates with the activation of substrates facilitated by energy transfer from photocatalysts, followed by the formation of a cyclic structure via the addition and coupling of free radicals. Notably, the stereochemistry is efficiently regulated using chiral phosphoric acid, thereby achieving products with remarkable enantioselectivity and diastereoselectivity. Remarkably, this methodology demonstrates versatility in reactivity, capably accommodating diverse olefin structures. It also displays an adept control over stereoselectivity for multi-substituted non-conjugated olefins and enols, presenting a pioneering and environmentally friendly synthetic route to chiral endocyclic compounds. Overall, this study constitutes a potentially substantial advancement in the realms of organic synthesis, photocatalysis, and asymmetric chemistry. Their findings introduce a more eco-friendly and efficient strategy for synthesizing novel molecules, aligning well with the burgeoning trend of sustainable chemical practices. I highly endorse the publication of this study in Nature Communications after the authors have addressed the minor revisions mentioned below.

(1) In the section on reaction condition optimization, the author acknowledges a certain degree of background reaction occurring in the absence of CPA (chiral phosphoric acid). Have the authors delved deeper into characterizing the by-products resulting from this set of reactions? This would provide compelling evidence that the introduction of the CPA catalyst not only governs the stereoselectivity but also significantly enhances the chemical selectivity of the process.

(2) During the assessment of different photocatalyst types, it was observed that photocatalysts possessing energy levels lower than or comparable to the triplet state of anthracene are unable to effectively trigger the reaction. However, for DPZ, 4-CzIPN, and iridium-based photocatalysts, there seems to be a discernible non-linear correlation between their use and the achieved reaction yields. The author should clarify the rationale behind this intriguing phenomenon.

(3) The manuscript would benefit from specifying the wavelength of the light source employed during the photo-induced reactions.

(4) From the NMR analysis, it appears that tests were conducted across multiple instruments, evidenced by varying ^{19}F NMR frequencies. The author should explicitly mention the different instrument models used under the General Information section. Moreover, if measurable, it is crucial to provide the coupling constants between fluorine atoms and carbon atoms.

(5) For enhanced clarity, Table 1 should include the full names corresponding to all abbreviations such as ppy, among others.

Reviewer #2:

Remarks to the Author:

In this manuscript, Jiang and co-workers describes a very interesting dearomative and highly enantioselective [4 + 2] photocycloaddition of anthracenes and alkenylazaarenes. While visible light-promoted [4+2] cycloaddition reactions via an energy transfer mechanism have been well-established, to the best of my knowledge, asymmetric [4+2] photocycloaddition reaction has yet to be achieved, presumably due to the challenges in inhibiting the very fast background reactions. In this work, the authors utilized DPZ, a photosensitizer developed by the authors' own group, as the energy transfer catalyst and SPINOL-derived phosphoramidate or phosphoric acid as the chiral catalyst to tackle this long-standing challenge. While DPZ has been widely utilized for single-electron transfer events, it has

rarely been used as an energy transfer catalyst. The [4 + 2] cycloaddition reaction exhibits very broad scope (more than 60 examples) and the products are generally obtained in good yields with high enantiopurity. Synthetic applications of the products have also been demonstrated and the proposed mechanism is reasonable based on the mechanistic studies.

Overall, this work represents a breakthrough in the field of photocycloaddition reactions. The novelty of this highly enantioselective system and the high efficiency in constructing otherwise inaccessible bridged skeleton render the current manuscript of significant interest to the readership of Nature Communications. I strongly recommend publishing this work after addressing the following minor additions/corrections.

- 1) Did the authors observe dimerization of anthracenes under the reaction conditions? I may miss it but it would be helpful if this point can be clarified.
- 2) Anthracen-9-ylmethanol (1f) was utilized as a substrate for the reaction shown in Table 3 and the authors indicated that the hydroxy group is very important for the reactivity as the corresponding OMe-substituted substrate is unreactive. However, this substrate is reactive in the reaction shown in Table 2 and provided product 3x in good yield with high enantioselectivity, can the authors briefly comment on this discrepancy?
- 3) The nitrogen atom of the alkenylazaarene seems to be very important, what if the N is placed at the C3 or C4 position of the vinylarenes (e.g., 3-vinylpyridine or 4-vinylpyridine derivatives)?
- 4) Several errors/typos are found in the SI. For example, on Page S14: "These data result in a linear Stern-Volmer relationship, consistent with the proposed mechanism in which the photoexcited catalyst is quenched by substrates" this sentence should be revised. Page S17, "+ ET0-0" should be "- ET0-0" in the second equation.

Reviewer #3:

Remarks to the Author:

The [4+2] cycloaddition is an efficient approach to assemble cyclohexane derivatives. Although recent photocatalysis through energy transfer has been explored to facilitate [4+2] cycloaddition radical reactions, the related enantiocontrol remains a challenge. In this manuscript, Jiang et al. realized the catalytic asymmetric [4+2] dearomative photocycloadditions of anthracene and its derivatives with alkenylazaarenes using dual catalyst system comprising DPZ photosensitizer and chiral Brønsted acid. Using this methodology, they have synthesized a wide range of pharmaceutical valuable cycloadducts with different azaarenes in high yields, ees and drs. This approach can also be used to construct distinct stereocenters, including all-carbon quaternary stereocenters and diversiform two adjacent stereocenters. This paper is well organized and written, the referee would recommend this paper to be accepted by Nature Communications after minor revisions.

Other comments:

- 1) On page 9 Table 2, the authors have used different solvents and different BA for various substrates to ensure good yields and stereoselectivities. Can the authors give some explanation or experience on the choice of ligands and solvents.
- 2) On page 11 Table 3, the authors have tried various vinylazaarenes, how about vinylazaarenes when R is alkyl?
- 3) On page 7 line 4, "The results clearly show that the photosensitizers with higher triplet energy (entries 4-6), but not lower (entries 2-3), than 1a can allow for a smooth conversion." Here ET of Rose Bengal (entry 3) is not lower than 1a.
- 4) On page 13 Fig. 2B, the proposed mechanism from 13 to 3a is too simple.
- 5) On page 6 Table 1, the expression for R in 4CzIPN should be revised.
- 6) On page 4 Fig 1C, substituent R of [4+4] product was missing. "et al." should be "etc."
- 7) In SI, 300MHz 1H NMR, 75MHz 13C NMR, and 565MHz for 19F NMR? Please check it carefully.

8) In SI page S5 Table S1, C8: Ar=2-ethyl or 2-ethylPh?

9) In SI page S55, the expression "methylene chloride-d2" and "CD2Cl2" should be unified.

Point to Point Response to Reviewers' Comments

Please take note that all the descriptive, positive comments of the reviews are omitted, and only reviewer's comments expressing their concerns/suggestions are listed below, which are followed by our response.

Reviewer 1' comments and our responses

1) In the section on reaction condition optimization, the author acknowledges a certain degree of background reaction occurring in the absence of CPA (chiral phosphoric acid). Have the authors delved deeper into characterizing the by-products resulting from this set of reactions? This would provide compelling evidence that the introduction of the CPA catalyst not only governs the stereoselectivity but also significantly enhances the chemical selectivity of the process.

RE: We are grateful for this important question. In fact, when without the CPA catalyst, the reactivity becomes extremely sluggish. Meanwhile, roughly no other side products, such as [4 + 4] cycloaddition adducts (note: not detected by HRMS analysis), were observed. In this context, the CPA catalyst is effective to improve the reactivity, therefore possessing a chance to provide enantiocontrol. To make it more intelligible to the reader, two sentences have been added in the revised manuscript (the first paragraph, page 7): (1) originating from the considerably worse chemical conversion; (2) It is worth mentioning that no [4 + 4] photocycloaddition product of **1a** was detected in these reactions, suggesting that the current photocatalysis platform is not suitable for such a competitive transformation.

2) During the assessment of different photocatalyst types, it was observed that photocatalysts possessing energy levels lower than or comparable to the triplet state of anthracene are unable to effectively trigger the reaction. However, for DPZ, 4-CzIPN, and iridium-based photocatalysts, there seems to be a discernible non-linear correlation between their use and the achieved reaction yields. The author should clarify the rationale behind this intriguing phenomenon.

RE: Many thanks! The following sentences have therefore been added in the revised manuscript (the first paragraph, page 7): It should be noted that only 45% yield of **3a** was obtained in the presence of Ir(ppy)₃ as a photocatalyst, which originates from poor reactivity, most likely due to insufficient energy transfer efficiency through molecular collisions (entry 5). Indeed, when extending the reaction time to 96 h, the transformation could be completed.

3) The manuscript would benefit from specifying the wavelength of the light source employed during the photo-induced reactions.

RE: The used light source has been introduced in the revised SI (Fig. S3, page S12).

4) From the NMR analysis, it appears that tests were conducted across multiple instruments, evidenced by varying ¹⁹F NMR frequencies. The author should explicitly mention the different instrument models used under the General Information section. Moreover, if measurable, it is crucial to provide the coupling constants between fluorine atoms and carbon atoms.

RE: Done. Thanks!

- 5) For enhanced clarity, Table 1 should include the full names corresponding to all abbreviations such as ppy, among others.

RE: Done. Thanks!

Reviewer 2' comments and our responses

- 1) Did the authors observe dimerization of anthracenes under the reaction conditions? I may miss it but it would be helpful if this point can be clarified.

RE: Actually, no [4 + 4] cycloaddition products from anthracenes were observed in the survey. On the basis of such helpful comments, the sentence 'It is worth mentioning that no [4 + 4] photocycloaddition product of **1a** was detected in these reactions, suggesting that the current photocatalysis platform is not suitable for such a competitive transformation.' has been added in the revised manuscript (the first paragraph, page 7).

- 2) Anthracen-9-ylmethanol (**1f**) was utilized as a substrate for the reaction shown in Table 3 and the authors indicated that the hydroxy group is very important for the reactivity as the corresponding OMe-substituted substrate is unreactive. However, this substrate is reactive in the reaction shown in Table 2 and provided product **3x** in good yield with high enantioselectivity, can the authors briefly comment on this discrepancy?

RE: Many thanks for this important suggestion. The sentence (the first paragraph, page 11) has been revised as: Nonetheless, we considered that this might stem from the ability of the hydroxyl group to provide an H-bonding donor to interact with O=P of the chiral catalyst, slightly reducing the activation energy to trigger these poorer reactive transformations originating from the higher steric hindrance of **4** compared to terminal olefins **2** and the weaker electron-deficient nature of **4** compared to 1,3-diketones **6** (*vide infra*).

- 3) The nitrogen atom of the alkenylazaarene seems to be very important, what if the N is placed at the C3 or C4 position of the vinylarenes (e.g., 3-vinylpyridine or 4-vinylpyridine derivatives)?

RE: According to the suggestions, reactions of anthracene with 3-vinylpyridine and 4-vinylpyridine under the established conditions have been performed. With respect to 3-vinylpyridine, the corresponding adduct was obtained in 13% yield with 9% ee. With respect to the latter olefin, the cycloadduct was achieved in 21% yield with 20% ee. Correspondingly, these olefins are less reactive than 2-vinylpyridine.

- 4) Several errors/typos are found in the SI. For example, on Page S14: "These data result in a linear Stern-Volmer relationship, consistent with the proposed mechanism in which the photoexcited catalyst is quenched by substrates" this sentence should be revised.

RE: Many thanks! The sentence has been corrected as: These data result in a linear Stern - Volmer relationship, consistent with the proposed mechanism in which the photoexcited catalyst is not quenched by substrates. In addition, in the second equation, "+ ETo-o" has been revised as "- ETo-o" (page S18).

Reviewer 3' comments and our responses

- 1) On page 9 Table 2, the authors have used different solvents and different BA for various substrates to ensure good yields and stereoselectivities. Can the authors give some explanation or experience on the choice of ligands and solvents.

RE: The sentence ‘It is worth mentioning that careful modulation of the reaction parameters, including chiral catalyst, temperature and solvent, is required to achieve high enantioselectivity, highlighting the elusive challenge of enantioselectivity in such photocycloaddition reactions, which are highly sensitive to these factors.’ has been modulated (page 9, the last sentence in the revised manuscript).

- 2) On page 11 Table 3, the authors have tried various vinylazaarenes, how about vinylazaarenes when R is alkyl?

RE: We are grateful for this important comment. We have tested a variety of β -alkyl substituted vinylazaarenes, but no reaction has been observed. A sentence ‘However, no reaction was observed for the β -alkyl substituted vinylazaarenes, revealing a remaining challenge.’ has therefore added in the revised manuscript (first paragraph, page 11).

- 3) On page 7 line 4, “The results clearly show that the photosensitizers with higher triplet energy (entries 4-6), but not lower (entries 2-3), than **1a** can allow for a smooth conversion.” Here ET of Rose Bengal (entry 3) is not lower than **1a**.

RE: We apologize for this error. In fact, the triplet energy of Rose Bengal is 40.9 kcal/mol, which is lower than that of **1a** (41.5 kcal/mol). We have checked the triplet energies for all species described in the manuscript, and errors have been corrected (note: a publication on the triplet energy of RB and Ir(ppy)₃ has been added as reference 33).

- 4) On page 13 Fig. 2B, the proposed mechanism from **13** to **3a** is too simple.

RE: Many thanks! Figure 2B has been revised accordingly.

- 5) On page 6 Table 1, the expression for R in **4CzIPN** should be revised.

RE: Done. Many thanks!

- 6) On page 4 Fig 1C, substituent R of [4+4] product was missing. “et al.” should be “etc.”

RE: Done. Many thanks!

- 7) In SI, 300MHz ¹H NMR, 75MHz ¹³C NMR, and 565MHz for ¹⁹F NMR? Please check it carefully.

RE: Many thanks! In addition to **3d**, the ¹⁹F NMR spectra of other products were examined using 600 M HZ NMR machine. We have examined all the Hz data for different NMR spectra.

- 8) In SI page S5 Table S1, **C8**: Ar=2-ethyl or 2-ethylPh?

RE: Many thanks! Ar = 2-ethylPh, and the error has been corrected.

- 9) In SI page S55, the expression “methylene chloride-d₂” and “CD₂Cl₂” should be unified.

RE: Done. Many thanks!

Sincerely,

Zhiyong Jiang

Reviewers' Comments:

Reviewer #1:

Remarks to the Author:

The authors have carefully addressed all the questions and concerns from the reviewers. The manuscript can be accepted as it is.

Reviewer #2:

Remarks to the Author:

The authors have nicely addressed all my previous concerns and I recommend acceptance of the manuscript as is. Congratulations to the authors!

Reviewer #3:

Remarks to the Author:

In the second revision, the authors have made some explanations and corrections. Although the explanation or experience on the choice of ligands and solvents are still ambiguous, this revision has addressed the main issues of this manuscript. Hence, this work now is recommended for acceptance in Nature Communications.